# Non-Hebbian plasticity transforms transient experiences into lasting memories

Islam Faress[1,2,3,4], Valentina Khalil[1,3,4†], Wen-Hsien Hou[2†], Andrea Moreno[1,3,4], Niels Andersen[1,3,4], Rosalina Fonseca[5], Joaquin Piriz[6‡], Marco Capogna[2,4§], Sadegh Nabavi[1,3,4]*

[1]Department of Molecular Biology and Genetics, Aarhus University, Aahrus, Denmark; [2]Department of Biomedicine, Aarhus University, Aarhus, Denmark; [3]DANDRITE, The Danish Research Institute of Translational Neuroscience, Aarhus University, Aahrus, Denmark; [4]Center for Proteins in Memory – PROMEMO, Danish National Research Foundation, Aarhus University, Aahrus, Denmark; [5]Cellular and Systems Neurobiology, Universidade Nova de Lisboa, Lisbon, Portugal; [6]Instituto de Fisiología Biología Molecular y Neurociencias (IFIBYNE), Universidad de Buenos Aires, Buenos Aires, Argentina

*For correspondence:
snabavi@dandrite.au.dk

†These authors contributed equally to this work

Present address: ‡Achucarro Basque Center for Neuroscience, Leioa, Spain

§Deceased

Competing interest: The authors declare that no competing interests exist.

**Abstract** The dominant models of learning and memory, such as Hebbian plasticity, propose that experiences are transformed into memories through input-specific synaptic plasticity at the time of learning. However, synaptic plasticity is neither strictly input-specific nor restricted to the time of its induction. The impact of such forms of non-Hebbian plasticity on memory has been difficult to test, and hence poorly understood. Here, we demonstrate that synaptic manipulations can deviate from the Hebbian model of learning, yet produce a lasting memory. First, we established a weak associative conditioning protocol in mice, where optogenetic stimulation of sensory thalamic input to the amygdala was paired with a footshock, but no detectable memory was formed. However, when the same input was potentiated minutes before or after, or even 24 hr later, the associative experience was converted into a lasting memory. Importantly, potentiating an independent input to the amygdala minutes but not 24 hr after the pairing produced a lasting memory. Thus, our findings suggest that the process of transformation of a transient experience into a memory is neither restricted to the time of the experience nor to the synapses triggered by it; instead, it can be influenced by past and future events.

## eLife assessment

This study presents **important** novel findings on how heterosynaptic plasticity can transform a weak associative memory into a stronger one, or produce a memory when stimuli were not paired. This work expands our views on the role of temporal- and input-specific plasticity in shaping learning and memory processes. The evidence, based on state-of-the-art in vivo manipulations, activity recordings, and behavioral analysis, is **convincing**. Findings will be of broad interest to neuroscience community, and especially those studying synaptic plasticity and associative memory.

## Introduction

Experience-dependent synaptic plasticity is widely regarded as the substrate of learning (*Kandel et al., 2016*; *Mayford et al., 2012*; *Squire and Kandel, 2009*), but see, (*Gallistel and King, 2009*;

*Gershman, 2023*). The dominant cellular model of learning, Hebbian plasticity, requires temporal and spatial specificity: the strength of a memory can be modified temporally only at the time of learning and spatially only at the encoding synapse and no other (*Malenka and Bear, 2004*; *Maxwell Cowan et al., 2003*). The most studied form of such plasticity is homo-synaptic long-term potentiation (homoLTP) of synaptic transmission or, as commonly known, LTP (*Malenka and Bear, 2004*; *Maxwell Cowan et al., 2003*). However, synaptic plasticity is neither temporally restricted to the time of the induction of the potentiation nor it is spatially confined to a single synapse (*Harvey and Svoboda, 2007*; *Koch, 2004*; *Stuart et al., 2016*; *Yuste, 2010*). Conceivably, therefore, the strength of a memory outside the time of learning can be modified by synaptic potentiation at the encoding synaptic input (homoLTP) or even in an independent input (heterosynaptic LTP, heteroLTP).

HeteroLTP has been identified in various synaptic pathways where a transient LTP can be stabilized by the induction of a more stable form of LTP on other synaptic inputs (*Fonseca, 2013*; *Frey et al., 2001*; *Frey and Morris, 1997*; *Shires et al., 2012*). HeteroLTP is not limited to the synapses that are already potentiated. In fact, heteroLTP can be induced in non-potentiated synapses as long as they receive subthreshold stimuli (*Harvey et al., 2008*; *Harvey and Svoboda, 2007*; *Hedrick et al., 2016*; *Murakoshi et al., 2011*). Subsequent studies established a temporal window, ranging from minutes to tens of minutes, within which heteroLTP can be induced (*Bear, 1997*; *Clopath et al., 2008*; *Govindarajan et al., 2006*; *Kastellakis et al., 2016*; *Kastellakis and Poirazi, 2019*; *O'Donnell and Sejnowski, 2014*; *Redondo and Morris, 2011*). Thus, an LTP protocol can produce synaptic potentiation at the stimulated synapses (Hebbian homoLTP), but it also modulates plasticity at other synapses that converge onto the same neuron (non-Hebbian heteroLTP). Consequently, the phenomenon of heteroLTP may accompany a homoLTP but remain undetected.

This has motivated us to examine the impact of LTP stimuli delivered to one set of synapses on memories formed by inputs to the same or a convergent set of synapses. Specifically, we asked if the two forms of plasticity (Hebbian homoLTP and non-Hebbian heteroLTP) differ in their efficacy in converting a transient experience to a lasting memory; and if the time window between the experience and the induction of plasticity influences the stabilization of the memory. In this work, we performed a side-by-side comparison between the Hebbian and non-Hebbian forms of LTP to answer these questions. We observed that the non-Hebbian form of plasticity which deviates from Hebbian rules is effective in stabilizing an otherwise transient aversive experience.

## Results
### Rationale for the approaches taken in this study
In general, to establish a causal link between changes in synaptic weight to the memory strength, we must fulfill a set of criteria. First, one must know which synapses encode the memory (*Stevens, 1998*). For this, it is necessary to probe the synaptic inputs whose strength can be measured and modified. One must further show that modifying these inputs produces a quantifiable behavioral readout (*Abdou et al., 2018*; *Jeong et al., 2021*; *Kim and Cho, 2017*; *Klavir et al., 2017*; *Nabavi et al., 2014*; *Roy et al., 2016*; *Zhou et al., 2017*). Additionally, to test the effect of heteroLTP, one must induce plasticity on an independent synaptic input that modifies the strength of the memory. This independent activation requires a means to selectively and independently activate the two synaptic inputs- a nontrivial task in an in vivo preparation (*Klapoetke et al., 2014*).

To investigate the temporal and spatial properties of non-Hebbian plasticity in relation to memory and behavior, we chose the defensive circuit in the lateral amygdala (*Fanselow and Poulos, 2005*; *Herry and Johansen, 2014*; *Janak and Tye, 2015*; *LeDoux, 2000*; *Maren and Quirk, 2004*; *Nabavi et al., 2014*; *Pape and Pare, 2010*; *Sah et al., 2008*; *Stevens, 1998*; *Tovote et al., 2015*). First, most of its excitatory neurons receive inputs from two sources, the thalamus and auditory/associative cortex (*Choi et al., 2021*; *Humeau et al., 2005*). Second, when these neurons receive a neutral conditioned stimulus (tone, CS) followed by an aversive unconditioned stimulus (shock, US), their synapses are potentiated to encode a memory of the aversive experience (conditioned response, CR) (*Fanselow and Poulos, 2005*; *Herry and Johansen, 2014*; *Janak and Tye, 2015*; *LeDoux, 2000*; *Maren and Quirk, 2004*; *Pape and Pare, 2010*; *Sah et al., 2008*; *Tovote et al., 2015*). To gain synapse-specific access to the CS input, we replaced a tone with optogenetic stimulation of the thalamic input (*Jeong et al., 2021*; *Kim and Cho, 2017*; *Nabavi et al., 2014*). This allowed precise control and monitoring

of the strength of the synaptic inputs encoding the memory (*Jeong et al., 2021*; *Kim and Cho, 2017*; *Nabavi et al., 2014*).

## Weak associative conditioning does not produce a lasting memory

The main objective of this work is to examine the efficacy of different forms of LTP in producing a lasting memory of an otherwise transient experience. Therefore, the memory under investigation must, by its nature, not be a lasting one. We have previously shown that an enduring CR can be produced by multiple pairs of optical co-activation of thalamic and auditory/associative cortical inputs with a footshock (*Nabavi et al., 2014*). We reasoned that reducing the number of pairings as well as

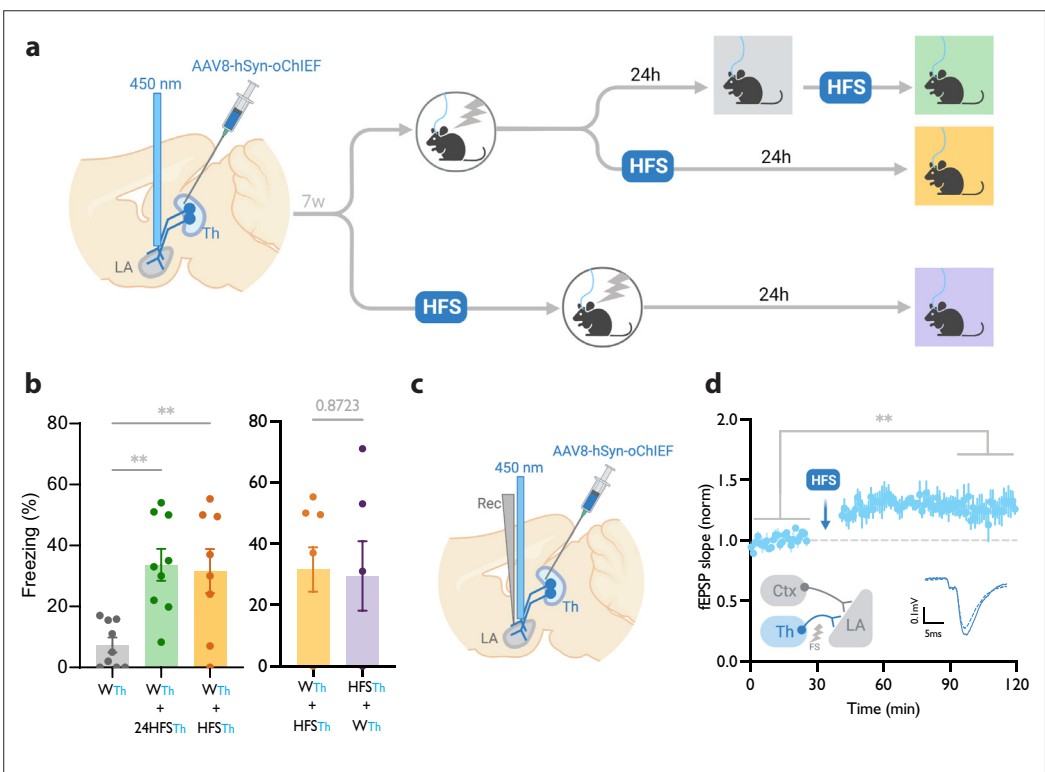

**Figure 1.** Homosynaptic long-term potentiation (LTP) stimulus minutes before, after, or 24 hr after a weak associative conditioning produces lasting memory. (**a**) Diagram showing the experimental timeline. (**b**) Left: High frequency stimulation (HFS) of the thalamic inputs (Th) to the lateral amygdala (LA) applied either 24 hr (WTh + 24hHFSTh, corresponding to panel a, top branch), or immediately after a weak thalamic associative conditioning (WTh + HFSTh, corresponding to panel a, middle branch), significantly increased the CS-evoked freezing (n=9; One-way ANOVA, FInteraction (2, 23)=8.202, p-value = 0.0020). Right: HFS of the thalamic input immediately before (HFSTh + WTh, corresponding to panel a, bottom branch) (n=6) or after a weak associative conditioning (n=8) (WTh + HFSTh, corresponding to panel a, middle branch) is equally effective in increasing the CS-evoked freezing. Colors of the bar graphs represent the experimental protocols for each group of mice (colored boxes in panel a). Subscripts with blue font indicate stimulation of the blue-shifted channelrhodopsin oChIEF using the selective procedure. (**c**) Diagram showing the experimental setup of the in vivo electrophysiology recording (Rec) in anesthetized mice. Evoked field EPSP was produced by blue light stimulation (450 nm) of thalamic inputs expressing oChIEF. (**d**) Plot of average in vivo field EPSP slope (normalized to baseline period) in the LA before and after HFS (n=5). Right inset: Superimposed traces of in vivo field responses to single optical stimulus before (dashed line) and after (solid line) HFS. Scale bar, 0.1 mV, 5 ms. Results are reported as mean ± S.E.M. **p<0.01. Ctx: Cortical input; Th: Thalamic input; LA: lateral amygdala; HFS: High Frequency Stimulation; EPSP: excitatory postsynaptic potential; WTh: Recall session after a weak thalamic associative conditioning.

The online version of this article includes the following figure supplement(s) for figure 1:

**Figure supplement 1.** Effect of optical thalamic conditioning protocol and modulation by homosynaptic LTP.

the duration of the footshock should result in a less robust CR. As will become clear later, here we must be able to produce a CR by using only one input. Therefore, we asked whether pairing optical activation of thalamic inputs alone with footshock can produce a lasting CR, and whether we can reduce the CR by using fewer pairings of CS and US, and with shorter US duration.

We injected an AAV virus expressing a fast, blue-shifted variant of channelrhodopsin, oChIEF (*Lin et al., 2013*), in the lateral thalamus. To optically activate the thalamic inputs to the LA, we implanted a fiber optic above the dorsal tip of the LA (*Figure 1a*, *Figure 1—figure supplement 1*). An optical CS alone did not produce a CR (*Figure 1—figure supplement 1b*), whereas temporal (but not non-temporal) multiple pairings of the optical CS with a footshock produced a freezing response (CR) measured 24 hrs later (60%±7), indicating the formation of a long-term associative memory (*Figure 1—figure supplement 1b*). Importantly, reducing the number of pairings with shorter US duration resulted in a significant reduction in the CR 24 hr following the conditioning (7%±2) (*Figure 1a and b*, *Figure 1—figure supplement 1b*).

## HomoLTP stimulus produces a lasting memory in weak associative conditioning

We next examined the efficacy of the LTP protocol in producing a long-term memory at different time points from the weak conditioning protocol. Delivering an optical LTP stimulus immediately before or after such a conditioning protocol on the same inputs (homoLTP) produced a lasting CR (*Figure 1b*). Remarkably, a homoLTP stimulus, even when delivered 24 hr after the conditioning, could produce a rapid CR comparable in magnitude to that obtained with immediate homoLTP. (*Figure 1a and b*). HomoLTP was as effective in mice that were tested prior to the induction protocol as those that were not (*Figure 1b*, *Figure 1—figure supplement 1d and e*). It is notable that a homoLTP stimulus in naïve animals failed to produce a CR (*Figure 1—figure supplement 1b*); whereas it did produce a CR as long as the animals received the conditioning protocol (*Figure 1b*).

To confirm that the optical homoLTP protocol was producing the expected effect on synaptic strength, we performed an in vivo recording from the LA in anesthetized mice expressing oChIEF in the thalamic inputs. Brief light pulses at the recording site produced in vivo field potentials which were potentiated by optical homoLTP stimulus (*Figure 1c and d*).

## Toward independent optical activation of thalamic and cortical inputs: Rendering a red-shifted channelrhodopsin insensitive to blue light

In addition to the thalamic inputs, most neurons within the LA receive direct projections from the cortical regions (auditory/associative) (*Choi et al., 2021*; *Humeau et al., 2005*). We, therefore, asked whether synaptic potentiation on the cortical inputs (heteroLTP) following the weak conditioning of thalamic inputs is effective in producing a long-term CR, as predicted by computational models (*O'Donnell and Sejnowski, 2014*).

Conceptually, the converging cortical and thalamic inputs to the LA can be activated independently using two opsins of distinct excitation spectra. The main obstacle is that all opsins, regardless of their preferred excitation spectrum, are activated by blue light (*Klapoetke et al., 2014*). Recent attempts addressed this problem by pairing blue-light sensitive anion channels with red-shifted ChR2, where red light derives action potentials, while blue light, through shunting inhibition, nullifies the effect of the red-shifted ChR2 (*Mermet-Joret et al., 2021*; *Vierock et al., 2021*). However, this approach, which is based on chloride influx, is not suitable for axonal terminal activation, where the chloride concentration is high (*Mahn et al., 2018*; *Mahn et al., 2016*).

A previous study demonstrated that prolonged illumination of axons expressing a red-shifted ChR2 reversibly renders the axons insensitive to further light excitation (*Hooks et al., 2015*). We, therefore, tested whether thalamic axons expressing ChrimsonR can become transiently non-responsive to blue light by the co-illumination with a yellow light. It must be noted that yellow light minimally activates the blue-shifted ChR2, oChIEF, (data not shown) the opsin that was later combined with ChrimsonR for independent optical activation of the thalamic and cortical axons. While activation of the thalamic axons expressing ChrimsonR by short pulses of blue light (10–15 mW) was effective in evoking a field potential, the light failed to produce a discernible response when the illumination coincided with a 500ms yellow light of sub milliwatt intensity. This was evident in whole-cell recording from slices (*Figure 2a–c*) as well as in vivo with single-pulse or high-frequency stimulation (*Figure 2d–h*).

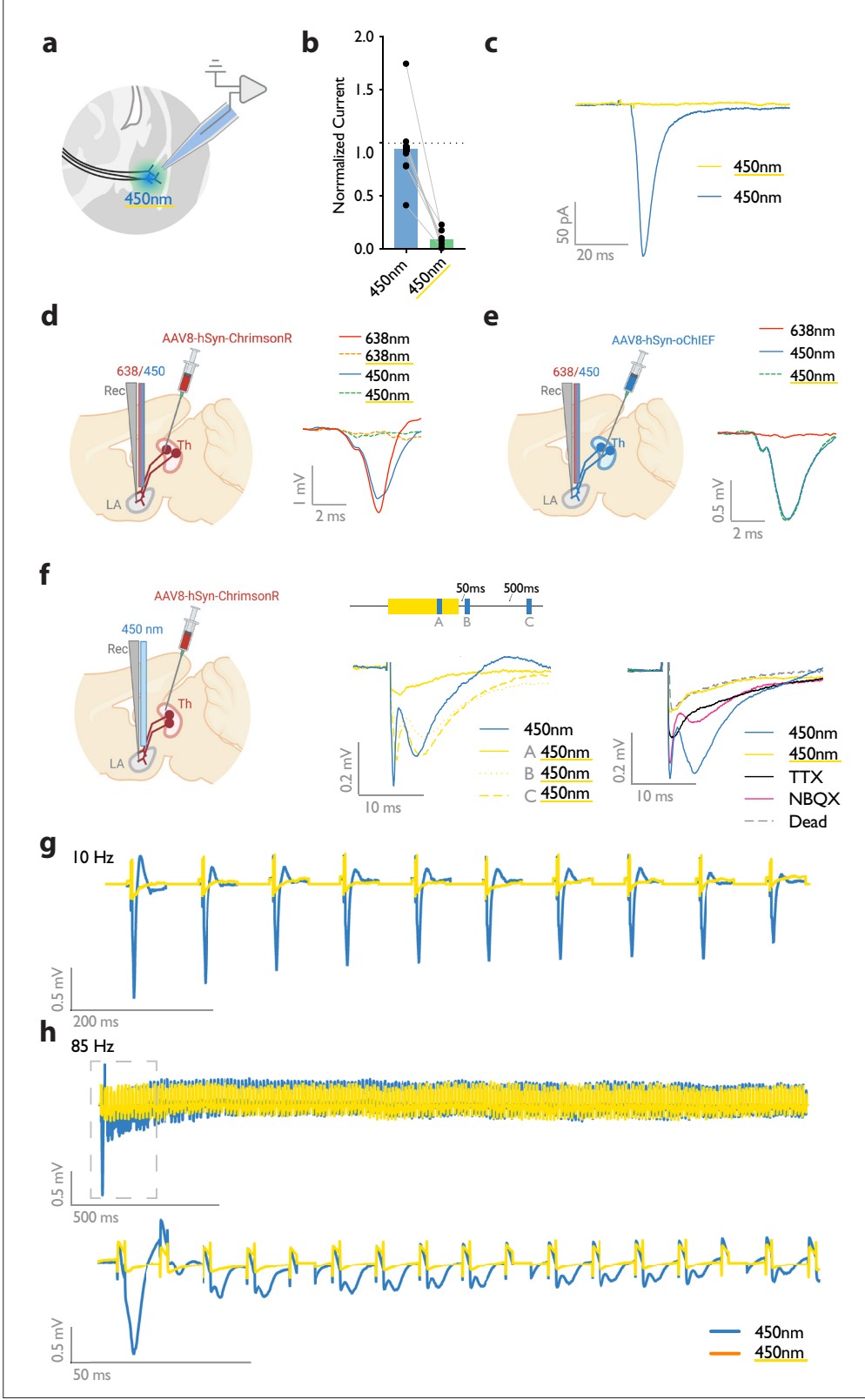

**Figure 2.** Submilliwatt yellow light renders a red-shifted channelrhodopsin insensitive to blue light. (**a**) Diagram showing ex vivo electrophysiology recordings in slices where ChrimsonR-expressing thalamic inputs to the lateral amygdala (black lines) were optically activated. Synaptic responses were evoked by pulses of 450 nm blue light (450 nm), or pulses of blue light co-illuminated with a 561 nm yellow light pulse (co-illumination). (**b, c**) Bar graph

*Figure 2 continued on next page*

*Figure 2 continued*

(normalized to blue light [450 nm] stimuli) (**b**), and example recording (scale bar, 50 pA, 20ms) (**c**), of optically driven synaptic responses to pulses of blue light (450 nm) or pulses of blue light co-illuminated with yellow light (450 nm with yellow underline). (**d**) Left: Diagram showing the experimental set up of electrophysiology recordings in freely moving mice where ChrimsonR-expressing thalamic inputs (Th) to the lateral amygdala (LA) were optically activated. Right: Comparison of a representative waveform average of the response to pulses of red light (638 nm), pulses of red light co-illuminated with a 500 ms yellow light pulse (638 nm with yellow underline), pulses of blue light (450 nm), and pulses of blue light co-illuminated with a 500 ms yellow light pulse (450 nm with yellow underline) (scale bar, 1 mV, 2 ms; n=3). (**e**) Left: Diagram showing the experimental set up of electrophysiology recordings in freely moving mice where oChIEF-expressing thalamic inputs (Th) to the lateral amygdala (LA) were optically activated. Right: Comparison of a representative waveform average of the response to pulses of red light (638 nm), pulses of blue light (450 nm), and pulses of blue light co-illuminated with a 500ms yellow light pulse (450 nm with yellow underline) (scale bar, 0.5 mV, 2 ms; n=3). (**f**) Left: Diagram showing the experimental setup of electrophysiology recordings in anesthetized mice where ChrimsonR-expressing thalamic inputs (Th) to the lateral amygdala (LA) were optically activated. Middle: Comparison of a representative waveform average of the response to pulses of blue light (450 nm), pulses of blue light co-illuminated with a 500 ms yellow light pulse (**A**), pulses of blue light following the yellow light pulse by 50 ms (**B**), or 500ms (**C**) (n=4). Right: Comparison of the waveform average responses to pulses of blue light (450 nm), pulses of blue light co-illuminated with a 500 ms yellow light pulse (450 nm with yellow underline), and pulses of blue light after sequentially applying NBQX and TTX, and later in a euthanized mouse (Dead), (n=4). Scale bar, 0.2 mV, 10 ms. (**g, h**) Representative traces for 10 Hz (**g**) and 85 Hz (**h**) stimulation of ChrimsonR-expressing thalamic inputs to the lateral amygdala, which were activated with blue light (450 nm, in blue). Yellow traces are the represantative evoked responses of the inputs to 10 Hz (**g**) and 85 Hz (**h**) blue light stimulation (450 nm) co-illuminated with a 561nm yellow light pulse (n=3).

With the co-illumination, fiber volley, and excitatory postsynaptic potential (the pre- and postsynaptic components, respectively) largely disappeared (*Figure 2f*). The responses gradually recovered to their original values within hundreds of milliseconds (*Figure 2f*). These data indicate that the observed insensitivity of ChrimsonR to blue light is more likely caused by the transient inactivation of the opsin rather than by the transmitter depletion or subthreshold depolarization of the axons. With an effective dual-color optical activation system at our disposal, we proceeded to investigate the effect of heteroLTP on the memory strength.

## Immediate heteroLTP stimulus produces a lasting memory in weak associative conditioning

Mice were injected with AAV-ChrimsonR in the thalamic inputs and AAV-oChIEF in the cortical inputs to the LA (*Figure 3a and b*, *Figure 3—figure supplement 1a*). To optically activate either thalamic or cortical inputs, we implanted a fiber optic above the dorsal tip of the LA (*Figure 3—figure supplement 1a and b*). Within 5 min after weak conditioning on thalamic inputs, we delivered an optical LTP protocol on the cortical inputs (heterosynaptic LTP, heteroLTP), while blocking the activation of the thalamic inputs using the co-illumination. Mice were tested for their long-term memory retention 24 hr later (*Figure 3a and b*). Similar to homoLTP, the induction of heteroLTP protocol immediately after the weak conditioning produced a long-term CR (*Figure 3c*). In mice expressing opsin only in the thalamic inputs, the same manipulation failed to produce a CR (*Figure 3c*). This demonstrates that the observed CR is caused by the heteroLTP.

As shown in the previous section, the delivery of an LTP protocol on the conditioned input (homoLTP) strengthens the memory even if delivered with a 24 hr delay. We then investigated whether the heteroLTP protocol similarly maintains its efficacy when a long period has elapsed since conditioning. Mice expressing AAV-ChrimsonR and AAV-oChIEF in the thalamic and cortical inputs received a weak conditioning protocol, followed 24 hr later by an LTP protocol on the cortical inputs (heteroLTP) (*Figure 3c*). In this condition, heteroLTP protocol, in contrast to homoLTP protocol, failed to produce a significant CR (*Figures 1b and 3c*).

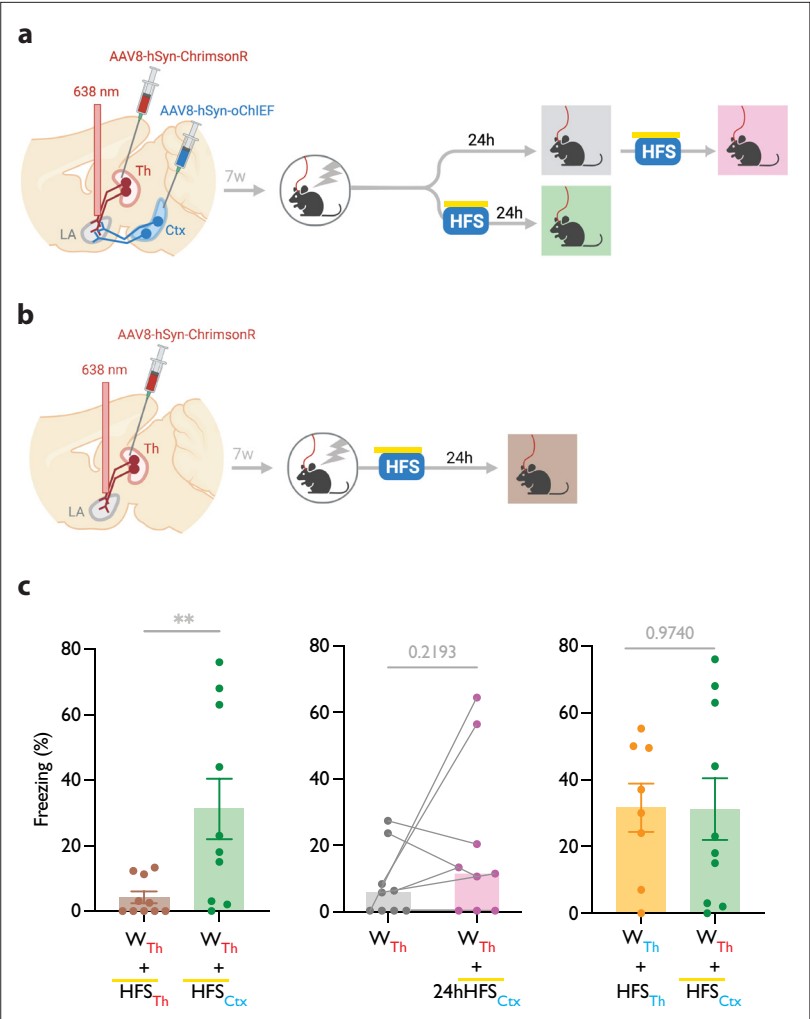

**Figure 3.** Heterosynaptic long-term potentiation (LTP) stimulus produces lasting memory if delivered within minutes after a weak associative conditioning. (**a, b**) Diagram showing the experimental timeline of the heterosynaptic LTP protocol manipulation following a weak thalamic associative conditioning. HFS with yellow upperline indicates that the delivery of high frequency stimulation with blue light overlapped with long pulses of yellow light. This co-illumination prevents the activation of ChrimsonR-expressing thalamic inputs (Th) by blue light, while the oChIEF-expressing cortical inputs remain unaffected. Note that yellow light specifically renders ChrimsonR, and not oChIEF, insensitive to blue light. (**c**) Left: High frequency stimulation (HFS) of the thalamic input expressing ChrimsonR immediately following a weak associative conditioning on the same input (WTh + HFSTh, corresponding to panel b) (n=9) was ineffective in producing the CS-evoked freezing. HFS with yellow upperline indicates that HFS with blue light overlapped with long pulses of yellow light. This was to prevent the activation of ChrimsonR-expressing thalamic inputs by blue light, as described above and detailed in *Figure 2*. The same HFS protocol in mice that additionally, expressed oChIEF in the cortical inputs (WTh + HFSCtx, corresponding to panel a, bottom branch) (n=10), significantly increased the CS-evoked freezing (heterosynaptic LTP) (Unpaired t-test, p-value = 0.0100). Middle: HFS on the cortical input, induced 24 hr after a weak associative conditioning (WTh + 24hHFSCtx, corresponding to panel a, top branch) was ineffective in producing the CS-evoked freezing. (n=9; Paired t-test, p-value = 0.2193). Right: Comparison of the effect of homosynaptic LTP protocol (WTh +HFSTh) (same dataset from *Figure 1b*) and heterosynaptic LTP protocol (WTh +HFSCtx) (same dataset from panel c, left) (Unpaired t-test, p-value = 0.9740). Results are reported as mean ± S.E.M. **p<0.01. Subscripts with red font and blue font indicate stimulation of the red-shifted channelrhodopsin ChrimsonR and the blue-shifted channelrhodopsin oChIEF, respectively.

The online version of this article includes the following figure supplement(s) for figure 3:

**Figure supplement 1.** Viral expression and optic fiber location in mice from *Figure 3*.

## Homo- and heteroLTP stimuli produce a lasting memory in unpaired conditioning

It has been shown that the thalamic→LA pathway, in addition to its role in the auditory-cued fear learning, is required for the formation of contextual fear memory (*Barsy et al., 2020*). This can be explained by the fact that the lateral thalamus, the thalamic gateway to the LA, collects signals from different brain regions of diverse modalities (*Barsy et al., 2020*; *Kang et al., 2022*; *Khalil et al., 2023*; *Ledoux et al., 1987*; *Linke et al., 1999*). We, therefore, asked if, in addition to cued associative conditioning, an LTP protocol can produce CR in an unpaired form of conditioning on the thalamic→LA pathway. First, we tested whether the thalamic inputs convey a footshock signal to the LA, which is a prerequisite for this paradigm. For this purpose, we took advantage of fiber photometry in freely moving mice. AAV virus expressing the genetically encoded Ca$^{2+}$ indicator GCaMP7s (*Dana et al., 2019*) was expressed in the thalamic inputs. GCaMP signal was collected through a fiber optic implanted above the tip of the LA (*Figure 4a*, *Figure 4—figure supplement 1a and b*). The time-locked GCaMP activity of the thalamic projections to the onset of the footshock was evident, demonstrating that the thalamic inputs convey the footshock signal to the LA (*Figure 4b*), confirming previous findings (*Barsy et al., 2020*). To further confirm this, we recorded the activity of the LA during footshock in mice with ablated lateral thalamus. This was done by the co-injection of AAV vectors expressing DIO-taCapsase3 and Cre recombinase in the lateral medial thalamus and GCaMP8m (*Zhang et al., 2023*) postsynaptically in the basolateral amygdala (*Figure 4c*, *Figure 4—figure supplement 1c and d*). The control group underwent the same procedure, but the thalamus was spared (no Cre-recombinase was injected). In the thalamic-lesioned mice, the footshock-evoked response in the LA was significantly reduced (*Figure 4d*). This further demonstrates that the aversive signal to the LA is conveyed largely through the thalamic inputs.

Next, we asked whether the induction of synaptic potentiation in this pathway following an unpaired conditioning, where footshock is not paired with the CS, would produce a long-term CR. It must be noted that previously we have shown that this protocol does not produce a detectable post-conditioning synaptic potentiation (*Nabavi et al., 2014*). Mice expressing AAV-oChIEF in the thalamic inputs received optical homoLTP stimulus on these inputs either immediately or 24 hr after the unpaired conditioning (*Figure 4e*). Immediate homoLTP stimulus, indeed, proved to be effective in producing a lasting CR even for the unpaired conditioning (*Figure 4f*); it is noteworthy that neither unpaired conditioning alone, nor optical homoLTP stimulus in naïve animals produced a CR (*Figure 4—figure supplement 2a*). HomoLTP protocol when delivered 24 hr later, produced an increase in freezing; however, the value was not statistically significant (*Figure 4—figure supplement 2b*). This observation is consistent with a previous report using only the unconditioned stimulus footshock (*Li et al., 2020*). This phenomenon is distinct from the paired form of conditioning which is receptive to homoLTP manipulation irrespective of the time of the delivery (*Figure 1b*).

Next, we investigated the behavioral consequence of heteroLTP stimulus on the unpaired conditioning. Mice expressing AAV-ChrimsonR in the thalamic and AAV-oChIEF in the cortical inputs received optical LTP stimulus on the cortical inputs immediately after footshocks (*Figure 4g*, *Figure 4—figure supplement 2c*). In this group, the heteroLTP protocol produced a CR, which was comparable in magnitude to the paired conditioned animals (compare *Figure 4h* with *Figure 3c*).

Based on this observation, we asked whether heteroLTP stimulus can induce potentiation in the thalamic synaptic inputs which were activated merely by footshock. Indeed, we observed that following footshocks, optical LTP delivery on the cortical inputs induced lasting potentiation on the thalamic pathway despite the fact that footshock on its own did not produce any detectable form of postsynaptic potentiation (*Figure 4i and j*). Without a footshock, high-frequency stimulation (HFS) of the cortical inputs did not induce synaptic potentiation on the thalamic pathway (*Figure 4—figure supplement 2d*). Therefore, although footshock on its own does not produce a detectable synaptic potentiation in thalamic inputs, it is required for heterosynaptic potentiation of this pathway.

## HeteroLTP stimulus produces lasting potentiation of synaptic inputs encoding memory in weak associative conditioning

We and others have shown that optical LTP protocols produce expected behavioral changes (*Nabavi et al., 2014*; *Roy et al., 2016*; *Zhou et al., 2017*), such as strengthening a memory (*Figures 1, 3 and 4*). However, we considered these approaches insufficient to establish a direct behavioral correlate

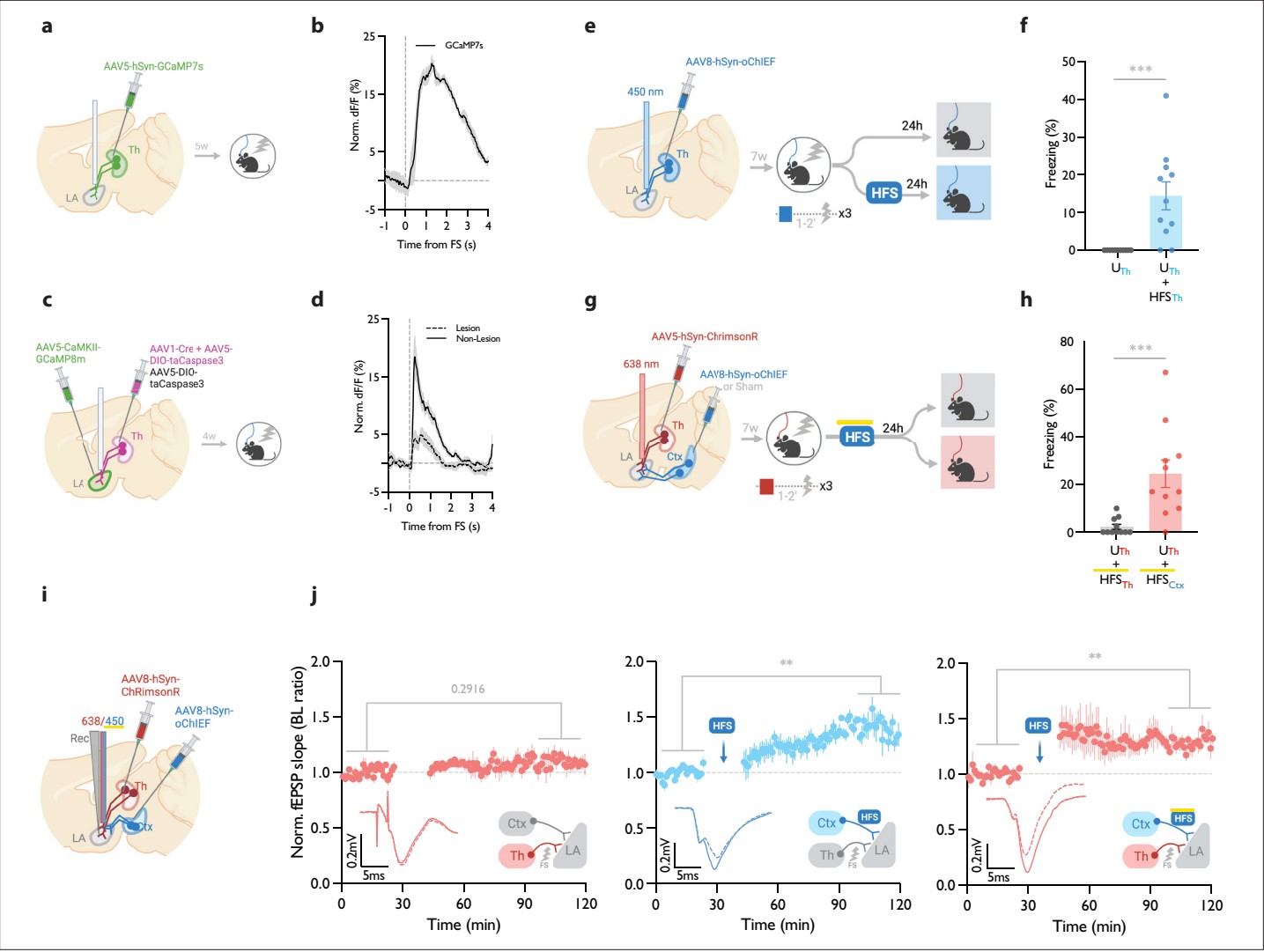

**Figure 4.** Homosynaptic and heterosynaptic long-term potentiation (LTP) protocols produce lasting memory when applied within minutes after a non-associative conditioning. (**a**) Diagram showing the experimental timelines for fiber photometry from thalamic inputs (Th) expressing GCaMP7s. (**b**) Averaged trace of the thalamic input activity in response to footshock (onset indicated by the dotted line), n=5. (**c**) Diagram showing the experimental timelines for fiber photometry from the lateral amygdala (LA) neurons expressing GCaMP8m with intact or lesioned thalamic inputs. (**d**) Averaged trace of the LA neurons activity in response to footshock (onset indicated by the dotted line) in mice with lesion (dash line) or no lesion (solid line) in the lateral thalamus (Th), n=6 per group. (**e**) Diagram showing the experimental timelines of the homosynaptic LTP protocol manipulation following an unpaired thalamic conditioning. (**f**) Unpaired conditioning on the thalamic inputs (UTh, corresponding to panel e, top branch) produced no CS-evoked freezing, while if unpaired conditioning was immediately followed by high frequency stimulation (HFS) on the same inputs (UTh + HFSTh, corresponding to panel e, bottom branch) it significantly increased the CS-evoked freezing (homosynaptic LTP), (n=11 per group; Mann-Whitney test, p-value = 0.0002). Subscripts with blue font indicate stimulation of the blue-shifted channelrhodopsin oChIEF using the selective procedure. (**g**) Diagram showing the experimental timelines of the heterosynaptic LTP protocol manipulation following an unpaired thalamic conditioning. (**h**) High frequency stimulation (HFS) of the thalamic input expressing red-shifted channelrhodopsin ChrimsonR immediately following an unpaired conditioning on the same input (UTh + HFSTh, corresponding to panel g, top branch) was ineffective in producing the CS-evoked freezing, while the same protocol in mice that, in addition, expressed oChIEF in the cortical inputs (UTh + HFSCtx, corresponding to panel g, bottom branch), significantly increased the CS-evoked freezing (heterosynaptic LTP) n=11 per group; Mann-Whitney test, p-value = 0.0002. During HFS, blue light pulses overlapped with long pulses of yellow light. This co-illumination prevents the activation of ChrimsonR-expressing thalamic inputs (Th) by blue light, while the oChIEF-expressing cortical inputs remain unaffected. Note that yellow light specifically renders ChrimsonR, and not oChIEF, insensitive to blue light. Subscripts with red font and blue font indicate stimulation of the red-shifted channelrhodopsin ChrimsonR and the blue-shifted channelrhodopsin oChIEF, respectively. (**i**) Diagram showing the experimental setup of the in vivo electrophysiology recordings (Rec) in anesthetized mice where the thalamic input expressing ChrimsonR and/or cortical input expressing oChIEF were optically activated independently. (**j**) Left: Plot of average in vivo field EPSP slope (normalized to baseline period) in LA evoked by optical activation of thalamic inputs, before and after footshock delivery (n=5; Paired t-test, p-value = 0.2916). Middle: Plot of average in vivo field EPSP slope (normalized to baseline period) in LA evoked by optical activation of cortical inputs (Ctx), before and

*Figure 4 continued on next page*

*Figure 4 continued*

after high frequency stimulation (HFS) of these inputs (n=6; Paired t-test, p-value = 0.0031). Right: Plot of average in vivo field EPSP slope (normalized to baseline period) in LA evoked by optical activation of thalamic inputs (Th), before and after HFS delivery on the cortical inputs (heterosynaptic LTP) (n=5; Paired t-test, p-value = 0.0074). HFS with yellow upperline indicates that the delivery of high frequency stimulation with blue light overlapped with long pulses of yellow light. This co-illumination prevents the activation of ChrimsonR-expressing thalamic inputs (Th) by blue light, while the oChIEF-expressing cortical inputs remain unaffected. Superimposed traces of in vivo field response to single optical stimulus before (dash line) and after (solid line) the induced protocols. Results are reported as mean ± S.E.M. **p<0.01; ***p<0.001. Scale bars, 0.2 mV, 5 ms.

The online version of this article includes the following figure supplement(s) for figure 4:

**Figure supplement 1.** Viral expression and optic fiber location in mice from *Figure 4a and c*.

**Figure supplement 2.** Effect of homosynaptic HFS on unpaired conditioning and unprimed heterosynaptic HFS.

of synaptic changes. To determine if synaptic potentiation accompanies increased fear response following heteroLTP induction, we resorted to in vivo recording in freely moving mice. We expressed AAV-ChrimsonR in the thalamic inputs, and AAV-oChIEF in the cortical inputs (*Figure 5—figure supplement 1a*). Six weeks after the injection, a customized optrode was implanted in the LA, which allows for the stimulation of the thalamic and cortical inputs as well as the measurement of the optically evoked field potential (*Figure 5a*, *Figure 5—figure supplement 1b*).

The baseline for the evoked field potential and the input-output curve of both pathways were recorded prior to the conditioning (*Figure 5—figure supplement 1c*). Blue light pulses produced smaller evoked responses when coincided with submilliwatt-long pulses of yellow light (data not shown). This further supports the efficacy of the dual optical activation approach that we adopted (*Figure 2*), which permits independent activation of the converging thalamic and cortical inputs in behaving mice. To induce a weak conditioning protocol on the thalamic inputs, mice received red light stimulation co-terminated with a footshock (*Figure 5a*). Within 5 min, we delivered an optical LTP protocol on the cortical inputs, while blocking the activation of the thalamic inputs using the co-illumination.

On the following day (recall day), we recorded evoked field responses prior to the memory retrieval. We observed a left-shifted input-output curve as well as lasting potentiated field responses in both thalamic and cortical pathways (*Figure 5b*, *Figure 5—figure supplement 1c*). Fifteen minutes later, mice were moved to a new context and tested for their memory recall by activating their thalamic inputs. Mice showed significantly increased freezing response during optical stimulation (*Figure 5c*). A weak conditioning protocol that was not followed by an optical LTP protocol on the cortical inputs failed to produce synaptic potentiation of the thalamic inputs (tested 2 hr and 24 hr after the LTP protocol; *Figure 5—figure supplement 1d and e*).

## HeteroLTP stimulus stabilizes a decaying form of synaptic potentiation in slices

Up to this point, we have shown that a memory and the underlying synaptic weight can be strengthened by the induction of LTP on an independent pathway. However, the notion of change in synaptic strength using an independent pathway was originally observed in slices (*Fonseca, 2013*; *Frey and Morris, 1997*). We, therefore, tested if the two pathways which we used for our behavioral manipulations can undergo similar changes in synaptic weight in a slice preparation where we have a more precise control on the activation and monitoring of synaptic plasticity. Stimulation of the thalamic inputs with a weak induction protocol (*Figure 5d*) resulted in a transient form of potentiation that regressed to the baseline within 90 min (*Figure 5e*). However, when the weak conditioning protocol was followed by a strong conditioning protocol on the converging cortical inputs, it produced a stable form of potentiation that lasted for the entire duration of the recording (*Figure 5e and f*).

## Discussion

Numerous forms of synaptic plasticity, such as long-term potentiation (LTP) have been described but their relation to long-term memory is poorly understood. Here, we investigated the temporal and

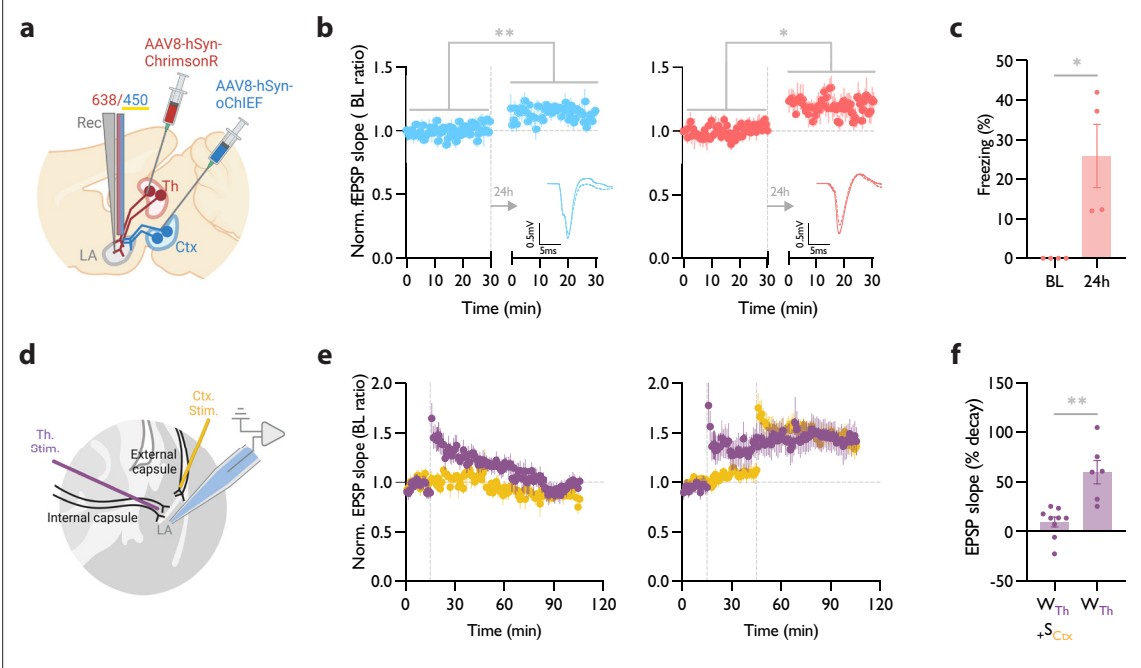

**Figure 5.** Heterosynaptic long-term potentiation (LTP) protocol when applied within minutes after a weak associative conditioning produces a long-lasting memory accompanied by the synaptic potentiation of the conditioned inputs. (**a**) Diagram showing the experimental setup of the in vivo electrophysiology recordings (Rec) in freely moving mice where the thalamic input expressing ChrimsonR and cortical input expressing oChIEF were optically activated independently. (**b**) Left: Plot of average in vivo field EPSP slope (normalized to baseline period) in LA evoked by optical activation of cortical inputs, before and 24 hr after a weak thalamic conditioning followed immediately by high-frequency stimulation (HFS) delivery on the same cortical inputs. Right: as left, except field EPSP was evoked by activation of thalamic inputs. The potentiation of the field EPSP of cortical (homosynaptic LTP) (n=4; Paired t-test, p=0.0082) as well as thalamic inputs (heterosynaptic LTP) (n=4; Paired t-test, p=0.0336) is evident 24 hr after the delivery of HFS. Superimposed traces of in vivo field response to single optical stimulus before (dash line) and after (solid line) HFS (Scale bars, 0.5 mV, 5 ms). (**c**) The behavioral responses of the mice tested for their homo- and hetero-synaptic plasticity in panel b. Note a significant CS-evoked freezing 24 hr after a weak thalamic conditioning followed immediately by HFS delivery on the cortical inputs (heterosynaptic LTP). These mice did not show a CS-evoked freezing prior to the protocol (BL) (n=4; Paired t-test, p=0.0478). (**d**) Positioning of the stimulating electrodes (Th. Stim and Ctx. Stim.) and the recording electrode. (**e**) (left) Weak stimulation of the thalamic input (purple circle) results in a transient LTP. No change was observed in the basal synaptic transmission of the cortical input (control pathway, yellow circle). Strong stimulation of the cortical input following the weak stimulation of the thalamic input stabilized synaptic potentiation of the thalamic input (right). Dash line indicates the onset of HFS induction. (**f**) A paired-comparison of the decay of synaptic potentiation of the thalamic input with (WTh + SCtx) or without (WTh) the strong stimulation of the cortical input. (10 animals, 15 slices; one cell per slice; Welch's t test, p=0.0062). Results are reported as mean ± S.E.M. *p<0.05; **p<0.01.

The online version of this article includes the following figure supplement(s) for figure 5:

**Figure supplement 1.** Histological verifications and in vivo freely-moving electrophysiology I/O curves.

input specificity learning rules by which Hebbian and non-Hebbian forms of synaptic potentiation modify the strength of a memory. We found that the strength of a memory can be enhanced by potentiating the synaptic inputs encoding that memory (homoLTP) prior to or after an aversive conditioning. Importantly, we show that potentiation of an independent synaptic input (heteroLTP) minutes after the conditioning is as effective in strengthening the memory.

Our in vivo electrophysiology recordings from freely moving animals showed a strong correlation between synaptic potentiation and the successful recall of the aversive memory, as late as 24 hr after the induction of heteroLTP; all the mice with the successful recall had a successful potentiation of the synaptic input (*Figure 5b and c*). This was accompanied by a lasting potentiation of the cortical input- the input that was used to induce heteroLTP in the thalamic inputs (*Figure 5b*). Such a lasting behavioral and electrophysiological consequence of heteroLTP has not been reported before.

The efficacy of heteroLTP stimulus when delivered 24 hr after the conditioning drops considerably, whereas homoLTP retains its capacity to strengthen the memory. These data are consistent with the

Synaptic Tagging and Capture (STC) model, which predicts that a heteroLTP protocol can stabilize a transient synaptic potentiation when induced minutes, but not hours prior to or after a weak LTP protocol (*Redondo and Morris, 2011*; *Rogerson et al., 2014*).

Perhaps the most surprising finding in this work is that homoLTP as well as heteroLTP effectively uncover an aversive memory in an unpaired conditioning paradigm; this form of conditioning on its own does not produce a detectable memory (*Figure 4—figure supplement 2a*). It is important to note that previously we have shown that unpaired conditioning not only fails to produce a CR, but also does not induce synaptic potentiation, as predicted by Hebbian models of plasticity (*Nabavi et al., 2014*). Similarly, in our in vivo recording, where anesthetized mice received multiple foot-shocks, no synaptic potentiation of thalamic inputs was detected (*Figure 4j*). The same protocol, however, when followed by heteroLTP stimulus, resulted in a synaptic potentiation that lasted for the entire duration of the recording (*Figure 4j*). This is not predicted by the STC model in which heteroLTP works only on the already potentiated synaptic inputs. In this respect, this phenomenon is more in line with the Cross Talk model, which predicts heteroLTP can result in potentiation of synapses that have undergone subthreshold stimulation but no potentiation (*Harvey et al., 2008*; *Harvey and Svoboda, 2007*). In the present context, the subthreshold activation could be the result of stimulation, and hence priming of the thalamic inputs by footshock. This is supported by the fact that in the absence of a footshock, LTP stimulus produces neither a CR nor a heterosynaptic poten-tiation (*Figure 4—figure supplement 2a and d*). As such, it appears that the mere potentiation of thalamic inputs is not sufficient to produce a memory, and some form of priming through associative or unpaired conditioning is essential.

Since we show that the thalamic inputs convey the footshock signal, the recovery of the CR following the LTP protocol on the same inputs (homoLTP) could be considered as a form of reinstatement, a well-known phenomenon where the mere presentation of a footshock after an extinguished CR rein-states the CR (*Shirah et al., 2016*; *Bouton and Bolles, 1979*). We think this is unlikely. First, we show that homoLTP is equally effective before the formation of the association. Additionally, we have shown previously that an LTP protocol is ineffective in restoring an extinguished CR (*Nabavi et al., 2014*).

It must be noted that computational models simulating a circuit with comparable pre- and post-synaptic layouts to ours yield similar results; that is heteroLTP stabilizes a weak memory. However, according to these models, heteroLTP in brain circuits with different pre- and postsynaptic arrange-ments, may produce different physiological and behavioral outcomes (*O'Donnell and Sejnowski, 2014*).

What cellular mechanisms could underlie the electrophysiological and behavioral phenomena we observed here? We consider some forms of postsynaptic intracellular diffusion from strongly stim-ulated cortical inputs to weakly stimulated neighboring thalamic inputs, as proposed by the Cross Talk and STC models. On the other hand, we consider the possibility of extracellular communication (*Engert and Bonhoeffer, 1997*) such as glutamate spillover to be unlikely. Extracellular communica-tion is mainly reported in the circuits at early developmental stages which lack a tight extracellular matrix sheath (*Asztely et al., 1997*). Additionally, as we have shown here (*Figure 4—figure supple-ment 2d*) and reported by others (*Doyère et al., 2003*), LTP induction on the cortical input produces no heterosynaptic effect on the naïve thalamic inputs. Taken together, our data point to an intracel-lular mechanism, which requires a prior priming but not necessarily a prior synaptic potentiation.

Consistent with this notion and complementary to our work, several studies have investigated the molecular and neuromodulatory mechanisms underlying the endurance of memories. For example, it has been shown that exposure to a novel experience strengthens memory encoding in appetitive and aversive learning paradigms (*Ballarini et al., 2009*; *Takeuchi et al., 2016*). Similarly, activation of dopaminergic inputs to the hippocampus after memory encoding enhances memory persistence, mimicking the effect of environmental novelty (*Rossato et al., 2009*; *Takeuchi et al., 2016*). De novo protein synthesis dependence and/or neuromodulator-signaling were suggested to be essential for this phenomenon. At this stage, we have no ground to speculate about the molecular mecha-nisms underlying our observations. Further studies are needed to reveal the molecular machinery that

enables non-Hebbian forms of plasticity that modify a memory and its synaptic strength across time and synapses.

# Materials and methods

**Key resources table**

| Reagent type (species) or resource | Designation | Source or reference | Identifiers | Additional information |
|---|---|---|---|---|
| Antibody | anti-NeuN antibody (Mouse Monoclonal) | Merk Millipore | MAB377 RRID: AB_2298772 | IF (1:500) |
| Antibody | Cy3 Goat anti-Mouse | Thermo Fisher Scientific | A10521 RRID: AB_1500665 | IF (1:500) |
| Other | DAPI | Sigma | D9542 | 5 µg/mL |
| Recombinant DNA reagent | AAV-8/2-hSyn1-oCHIEF_dTomato | VVF | V391-8 | |
| Recombinant DNA reagent | AAV-5/2-hSyn1-chI-jGCaMP7s | VVF | V406-5 | |
| Recombinant DNA reagent | AAV-5/2-mCaMKIIα-jGCaMP8m | VVF | V630-5 | |
| Recombinant DNA reagent | AAV-5/2-hSyn1-chI-ChrimsonR_tdTomato | VVF | V334-5 | |
| Chemical compound, drug | Fentanyl | Hameln | 007007 | |
| Chemical compound, drug | Midazolam | Hameln | 002124 | |
| Chemical compound, drug | Medetomidine | VM Pharma | 087896 | |
| Chemical compound, drug | IsoFlo vet 100% | Zoetis | 37071/4000 | |
| Chemical compound, drug | Urethane | Sigma | U2500 | |
| Chemical compound, drug | NBQX disodium salt hydrate | Sigma | N183 | |
| Chemical compound, drug | Tetrodotoxin citrate | Hellobio | HB1035 | |
| Software, algorithm | GraphPad Prism | GraphPad Software | Version 9 | |
| Software, algorithm | ImageJ | National Institutes of Health | 1.53t | |
| Software, algorithm | Doric Studio | Doric Lenses | 5.4.1.23 | |
| Software, algorithm | MATLAB | MathWorks, Inc. | R2021b | |
| Software, algorithm | Photometry-Signal-Analysis | This paper; *NabaviLab-Git, 2024* | https://github.com/NabaviLab-Git/Photometry-Signal-Analysis | Code used for the Fiber photometry analysis. |

## Animals

Male mice of the strain C67BL/6JRj were purchased from Janvier Labs, France. Mice are purchased at the age of 6–8 weeks. All mice were housed in 12 hr light/dark cycle at 23°C and had ad libitum food and water access. Mice were housed 4 per cage. All procedures that involved the use of mice were approved by the Danish Animal Experiment Inspectorate (permit numbers: 2020-15-0201-00421 and 2023-15-0201-01431).

## Viruses

Recombinant adeno-associated viral vectors (AAV) were purchased from the viral vector facility VVF, at the University of Zurich, Switzerland. Serotype 8,

AAV-2-hSyn1-oCHIEF_tdTomato(non-c.d.)-WPRE-SV40p(A) had physical titer of 6.6×10E12 vg/mL. Serotype 5, AAV-1/2-hSyn1-chI-ChrimsonR_tdTomato-WPRE-SV40p(A) had a physical titer of 5.3×10E12 vg/mL. Serotype 5, AAV-2-mCaMKIIα-jGCaMP8m-WPRE-bGHp(A) had a physical titer of 6.6×10E12 vg/mL. Serotype 5, AAV-2-hSyn1-chI-jGCaMP7s-WPRE-SV40p(A) had a physical titer of 7.7×10E12 vg/mL. Serotype 5, ssAAV-2-hEF1α-dlox-(pro)taCasp3_2 A_TEVp(rev)-dlox-WPRE-hGHp(A) had a physical titer of 4.7×10E12 vg/mL. Serotype 1, scAAV-1/2-hCMV-chI-Cre-SV40p(A) had physical titer of 1.0×10E13 vg/mL. Serotype 8, AAV-2-hSyn1-hM4D(Gi)_mCherry-WPRE-hGHp(A) had physical titer of 4.8×10E12 vg/mL.

## Surgery

Mice were 7–9 weeks at the time of stereotaxic surgery. Mice were anesthetized with isoflurane and maintained at 1% throughout the surgery in the stereotaxic setup (Kopf 940) and a heating pad maintained body temperature at 37°C. Viruses were injected with a volume of 500–700 nL over 3–4 min. Auditory/associative cortex coordinates (all in mm and from Bregma) are (–2.85 AP, –4.4 ML, and +1.6 DV (from the skull surface)). Lateral thalamus coordinates are (–3.15 AP, –1.85 ML, and +3.5 DV (from the skull surface)). LA coordinates are (–1.65 AP, –3.45 ML, and +3.45 DV (from the skull surface)). Optic fiber cannulas were cemented with dental cement, Superbond (SUN MEDICAL, Japan). All the injections and optic fiber implantations were performed in the right hemisphere.

## Optogenetics

ChR expressing AAVs were injected into the thalamic and the cortical regions projecting to the LA, and a 6–8 week expression time was given to allow for a high and stable expression in the axons. In freely moving mice, a 200 micrometer (Thorlabs 200 EMT, NA 0.39) optic fibers cannulae were implanted in the same surgery to target LA. The optic fiber cannulae were fabricated manually. The optic fiber was scored with an optic fiber scribe (Thorlabs s90 carbide scribe) and then pulled to break. Next, the optic fiber was inserted into the ferrule, and the output was measured with a power meter (Coherent Laser Check); 10 percent loss was the maximum allowed loss after coupling to the patch cord (Thorlabs 200 um NA 0.39). Afterward, the length was adjusted to 4 mm (the exposed optic fiber) and glued with a UV-curable glue. After gluing, the opposite end was scored and cut, and the output was measured again. The light output was confirmed to have a concentric-circle pattern.

In experiments with oChIEF, a 450 nm laser diode (Doric) was used with a light intensity of 10–15 mW. In the experiments with ChRimsonR, a 638 nm laser diode (Doric) was used with a light intensity of 10–15 mW, and a 561 nm laser diode (Vortran Laser Technology, USA) at the intensity of 1 mW for co-illumination when performing independent optical activation. All the freely moving experiments were done with a rotary joint (Doric Lenses, Canada). After each experiment, the verification of the brain stimulation location was performed after PFA fixation and slicing. For optimal optic fiber tract marking, the whole head of the mouse was left in 10% formalin for a week with agitation. Mice were excluded if the viral expression and/or the optic fiber locations were off-target.

## In vivo electrophysiology

Mice were anesthetized with Urethane, and Ethyl-Carbamate 2 mg/kg and placed in the stereotaxic setup, and a heating pad maintained the body temperature at 37°C. Multichannel system ME2100 was used for signal acquisition, and a Neuronexus opto-silicone probe with 32 channels was used to record the signal. Raw data were filtered (0.1–3000 Hz), amplified (100x), digitized, and stored (10 kHz sampling rate) for offline analysis with a tethered recording system (Multichannel Systems, Reutlingen, Germany). Analysis was performed using custom routines. The initial slope of field excitatory postsynaptic potentials was measured as described by *Nabavi et al., 2014*.

The light-evoked signal was recorded from the LA in the right hemisphere. For LTP experiments, a baseline of the light-evoked fEPSP was measured for at least 20 min or until it was stable at 0.033 Hz, 1–2 ms pulses. At the of the baseline recording, three mild foot shocks were delivered to the mouse at the same intervals and intensity as the behavioral protocol. Only mice in *Figure 4—figure supplement 2d* did not receive foot shocks. After the foot shock delivery, an HFS stimulation protocol was applied. The protocol consisted of 20 trains of 200 pulses of 2 ms, 450 nm light at 85 Hz with a 40 s inter-train interval. Immediately after the HFS, the light-evoked fEPSP was measured for at least

45 min to ensure the stability of the outcome of the LTP. This HFS protocol was used for all the experiments with HFS stimulation.

In the experiment that involved drug application, approximately 1 µL of the drug (TTX: 10 ng or NBQX: 1 µg) was applied onto the shank of the silicone probe and was inserted again. After each experiment, the brain recording location was verified through a stereoscope after PFA fixation and slicing.

For in vivo electrophysiological recordings from freely moving mice, a customized microdrive was designed to enable concurrent optical stimulation and recording of neuronal activity (modified from *Kvitsiani et al., 2013*). The microdrive was loaded with a single shuttle driving a bundle of three tetrodes (Sandvik) and one 200 µm-diameter optical fiber (Doric lenses). Three weeks after the virus injection, a microdrive was implanted. For this, mice were anesthetized with 0.5 mg/kg FMM composed of the following mixture: 0.05 mg/ml of fentanyl ([Hameln, 007007] 0.05 mg/kg), 5 mg/mL of midazolam ([Hameln, 002124] 5 mg/kg), and 1 mg/mL of medetomidine (VM Pharma, 087896). To target the LA in the right hemisphere, a ~1 mm diameter hole was drilled through the skull at the coordinates AP, −1.8 mm; ML, +3.4 mm. The microdrive was positioned with the help of a stereotaxic arm (Kopf Instruments) above the hole with protruding tetrodes. The optical fiber and tetrodes were gradually lowered to a depth of 500 µm from the brain surface. A screw electrode was placed above the cerebellum to serve as the reference and ground electrode. The microdrive was secured to the skull with ultraviolet light curable dental cement (Vitrebond Plus) followed by a layer of Superbond (SUN MEDICAL). Tetrodes and the optical fiber were lowered by a further 2500 µm before mice recovered from anesthesia. The post-operative analgesia Buprenorphine (0.1 mg/kg, S.C.) was administered 30 min before the end of surgery. Mice were allowed to recover for at least a week after the implantation.

Electrophysiological recordings were performed using a Neuralynx Cheetah 32 system. The electrical signal was sampled at 32 kHz and band-pass filtered between 0.1–8000 Hz.

## Ex vivo slice electrophysiology (related to Figure 2a–c)

Experimental procedures were approved by the Animal Care and Use Committee of the University of Buenos Aires (CICUAL). Briefly, 4- week-old C57 mice (n=4) were injected with 250 nl of AAV-1/2-hSyn1-chI-ChrimsonR_tdTomato-WPRE-SV40p at the MGN. After 3 weeks of expression, the animals were sacrificed and the brain was removed and cut into 300 µm coronal slices in a solution composed of (in mM): 92 N-Methyl-D-glucamine, 25 glucose, 30 NaHCO3, 2.5 KCl, 1.25 NaH2PO4, 20 HEPES, 2 Thiourea, 5 Na-ascobate, 3 Na-pyruvate, 10 MgCl2, and 0.5 CaCl2 (equilibrated to pH 7.4 with 95% O2–5% CO2); chilled at 4 °C. Slices containing the BLA were transferred to a 37 °C warmed chamber filled with the same solution and incubated for 10 min. After this period slices were transferred to a standard ACSF solution of composition (in mM): 126 NaCl, 26 NaHCO3, 2.5 KCl, 1.25 NaH2PO4, 2 MgSO4, 2 CaCl2, and 10 Glucose (pH 7.4), at room temperature. Recordings started 1 hr later and were performed in this same ACSF solution. Patch-clamp recordings were done under a microscope (Nikon) connected to a Mightex Illumination system for 470 nm, and 532 nm light delivery. Whole-cell patch-clamp recordings were done using a K-gluconate-based intracellular solution of the following composition (in mM): 130 K-gluconate, 5 KCl, 10 HEPES, 0.6 EGTA, 2.5 MgCl2·6H2O, 10 Phospho-creatine, 4 ATP-Mg, 0.4 GTP-Na3. Glutamatergic AMPA-mediated synaptic responses were recorded at –60 mV holding potential under blockage of GABAa and NMDA receptors (Picrotoxin 100 µM and APV 100 µM). Light stimulation consisted in 2ms pulses of 470 nm light at 10 mW, and co-illumination consisted of 450 ms of 532 nm light at 1 mW that co-terminated with stimulation light.

## Ex vivo slice electrophysiology (related to Figure 5d–f)

A total of 15 slices prepared from 10 Black6/J mice (3–5 weeks old) were used for electrophysiological recordings. All procedures were approved by the Portuguese Veterinary Office (Direcção Geral de Veterinária e Alimentação - DGAV). Coronal brain slices (300 µm) containing the lateral amygdala were prepared as described previously (*Fonseca, 2013*). Whole-cell current-clamp synaptic responses were recorded using glass electrodes (7–10 MΩ; Harvard apparatus, UK), filled with internal solution containing (in mM): K-gluconate 120, KCl 10, Hepes 15, Mg-ATP 3, Tris-GTP 0.3 Na-phosphocreatine 15, Creatine-Kinase 20 U/ml (adjusted to 7.25 pH with KOH, 290mOsm). Putative pyramidal cells were selected by assessing their firing properties in response to steps of current. Only cells that had a

resting potential of less than –60 mV without holding current were taken further into the recordings. Neurons were kept at –70 mV with a holding current below –0.25nA. In current clamp recordings, the series resistance was monitored throughout the experiment and ranged from 30 MΩ-40MΩ. Electrophysiological data were collected using an RK-400 amplifier (Bio-Logic, France) filtered at 1 kHz and digitized at 10 kHz using a Lab-PCI-6014 data acquisition board (National Instruments, Austin, TX), and stored on a PC. Offline data analysis was performed using a customized LabView-program (National Instruments, Austin, TX). To evoke synaptic EPSP, tungsten stimulating electrodes (Science Products, GmbH, Germany) were placed on afferent fibers from the internal capsule (thalamic input) and from the external capsule. Pathway independence was checked by applying two pulses with a 30 ms interval to either thalamic or cortical inputs and confirming the absence of crossed pair-pulse facilitation. EPSPs were recorded with a test pulse frequency for each individual pathway of 0.033 Hz. After 15 min of baseline, transient LTP was induced with a weak tetanic stimulation (25 pulses at a frequency of 100 Hz, repeated three times with an interval of 3 s) whereas long-lasting LTP was induced with a strong tetanic stimulation (25 pulses at a frequency of 100 Hz, repeated five times, with an interval of 3 s).

As a measure of synaptic strength, the initial slope of the evoked EPSPs was calculated and expressed as percent changes from the baseline mean. Error bars denote SEM values. For the statistical analysis, LTP values were averaged over 5 min data bins immediately after LTP induction (T Initial) and at the end of the recording (T Final 95–100 min). LTP decay was calculated by [(T Initial –T Final)/T Final*100].

## Fiber photometry

GCaMP fluorescent signal was acquired by a Doric fiber photometry system and through an optic fiber that is identical to the optogenetics ones described above. A pigtailed rotary joint (Doric) was used for all fiber photometry experiments in freely moving mice. Doric Lenses single site Fiber Photometry Systems with a standard 405/465 nm system fluorescent minicube ilFMC4-G2_E(460-490)_F(500-540)_O(580-680)_S. The 405 nm was modulated at 208.616 Hz, and 465 nm was modulated at 572.205 Hz through the LED module driver. When the fiber photometry experiments were combined with optogenetics and/or electrophysiology recordings, the 638 nm laser diode was used to deliver the red light. A TTL generator device (Master 9) was used to time-stamp the signals. The data was acquired through Doric Studio and analyzed in Doric studio and by a custom MatLab script. The code used for the analysis is freely available at the following link: https://github.com/NabaviLab-Git/Photometry-Signal-Analysis, copy archived by **NabaviLab-Git, 2024**. Briefly, the signals were downsampled to 120 Hz using local averaging. A first-order polynomial was fitted onto the data, using the least squares method. To calculate the relative change in fluorescence, the raw GCaMP signal was normalized using the fitted signal, according to the following equation: \deltaF/F = (GCaMP signal - fitted signal)/(fitted signal). Behavioral events of interest were extracted and standardized using the mean and standard deviation of the baseline period.

## Behavior

Eight weeks after the AAV injection, around 2 p.m., the mice were single-housed 20 min before the conditioning in identical cages to the home cages. Ugo Basile Aversive conditioning setup was used for all the experiments. The conditioning protocol was preceded by a pre-test, optical stimulation at 10 Hz for 30 s testing optical CS, identical to the one used in the 24 hr test. This step ensures that optical stimulation before conditioning and HFS does not cause any freezing or seizures. The strong conditioning protocol consisted of five pairings of a 2 s long optical CS at 10 hz, 20 pulses, co-terminated (last 15 pulses) with a 1.5 s foot shock 1 mA. The weak conditioning protocol was composed of three pairings of a 1.5 s long optical CS at 10 hz, 15 pulses, co-terminated (last 10 pulses) with a 1 s foot shock 1 mA. Twenty-four hours later, the mice were tested in a modified context with bedding on the context floor, and chamber lights switched off. The mice were given a 2 min baseline period or until they maintained a stable movement index and did not freeze at least 1 min before the delivery of the testing optical CS. The testing optical CS was delivered twice, 2 min apart. Freezing was automatically measured through Anymaze (Stoelting, Ireland; version 5.3). Freezing percentages indicated the time the mouse spent freezing (in the 2 CSs) divided by 60 and multiplied by 100. The unpaired conditioning had the same number of pairings and parameters of the optical CS and

the foot shock, as the weak conditioning protocol, with the difference that they were never paired, separated by 1–3 min. Depending on the experiment, the HFS protocols (described above) were either delivered in the conditioning chamber within 5 min from the beginning or at the end of the conditioning session, or in the testing chamber within 5 min from the end of the 24 hr recall. The control groups remained in the same context for the same amount of time as the mice that received the HFS protocol.

## Drugs

All drugs were dissolved in sterile PBS from stock solutions. NBQX at 50 micromolar (Sigma) and TTX 0.5 micromolar (HelloBio) were added to the silicone probe's shank (5 microlitres). NBQX was added before the TTX.

## Immunofluorescence

The mice were anesthetized with Isofluorane and euthanized by cervical dislocation. The heads were collected and stored for 7 days in 10% formalin at room temperature. Then, the brains were sliced into 100–120 µm thick slices in PBS on Leica Vibratome (VT1000 S).

To exclude any virus-mediated toxicity, the brains were stained for NeuN. Slices were permeabilized with PBS-Triton X 0.5% plus 10% of Normal Goat Serum (NGS; Thermo Fisher Scientific, 16210064) and blocked in 10% Bovine Goat Serum (BSA; Sigma, A9647) for 90 min at room temperature. Subsequently, the slices were incubated with anti-NeuN antibody mouse (Merk Millipore, MAB377; 1:500) in PBS-Triton X 0.3%, 1% NGS, and 5% BSA. The incubation lasted for 72 hr at 4 °C. At the end of the 72 hr incubation, the slices were washed three times in PBS at room temperature. Next, the slices were incubated in Cyanine 3 (Cy3) goat anti-mouse (Thermo Fisher Scientific, A10521, 1:500) in PBS-Triton X 0.3%, 1% NGS, and 5% BSA for 24 hr at 4 °C. Finally, nuclear staining was performed using 1:1000 of DAPI (Sigma, D9542) for 30 min at room temperature. Brain slices were mounted on poly-sine glass slides (Thermo Scientific) with coverslips (Housein) using Fluoromount G (Southern Biotech) as mounting media.

## Imaging

Imaging was performed by using a virtual slide scanner (Olympus VS120, Japan). Tile images were taken by the whole brain slides by using 10 X (UPLSAPO 2 10 x/0,40) or 20 X objective (UPLSAPO 20 x/0,75). The emission wavelength for Alexa 488 was 518 nm with 250 ms of exposure time. For Cy3, the emission wavelength was 565 nm with 250 ms of exposure time. The brain slices were visually inspected to confirm the virus expression in the thalamic and cortical regions projecting to the LA and to determine the optic fiber location in the LA.

## Statistics

Statistical analyses were done via Prism 8.01 (GraphPad Software, San Diego, CA, USA). All the data are represented as mean ± SEM. Before choosing the statistical test, a normality test (Shapiro-Wilk and D'Agostino-Pearson normality test) was done on all data sets. If the data presented a normal distribution, then a parametric test was used to calculate the statistical differences between groups. The statistical methods and the p-values are mentioned in the figure legends.

## Acknowledgements

We thank R Malinow and the current and previous members of the Nabavi laboratory for their suggestions during the progress of this project and their comments on the manuscript. We also thank R Morris, P Sterling, and J Lima for their comments on the manuscript. This study was supported by an ERC starting grant to SN (22736), by Lundbeck NIH Brain Initiative grants to SN (R360-2021-650 and R273-2017-179) by the Danish Research Institute of Translational Neuroscience to SN (19958), by PROMEMO (Center of Excellence for Proteins in Memory funded by the Danish National Research Foundation) to S.N (DNRF133). JP is supported by PICT-2021-I-A-00494 from the National Agency for Scientific and Technological Promotion of Argentina (ANPCyT). Figures were created with BioRender.com.

## Additional information

### Funding

| Funder | Grant reference number | Author |
| --- | --- | --- |
| European Research Council | 22736 | Sadegh Nabavi |
| Lundbeck Foundation | R360-2021-650 | Sadegh Nabavi |
| Lundbeck Foundation | R273-2017-179 | Sadegh Nabavi |
| Danmarks Grundforskningsfond | DNRF133 | Sadegh Nabavi |
| The National Agency for Scientific and Technological Promotion of Argentina | PICT-2021-I-A-00494 | Joaquin Piriz |
| Danish Research Institute of Translational Neuroscience | 19958 | Sadegh Nabavi |

The funders had no role in study design, data collection and interpretation, or the decision to submit the work for publication.

### Author contributions

Islam Faress, Data curation, Formal analysis, Validation, Investigation, Visualization, Methodology, Writing – original draft, Writing - review and editing; Valentina Khalil, Data curation, Formal analysis, Validation, Investigation, Visualization, Methodology, Writing – original draft; Wen-Hsien Hou, Joaquin Piriz, Formal analysis, Investigation, Visualization, Methodology; Andrea Moreno, Formal analysis, Investigation, Visualization, Methodology, Writing – original draft; Niels Andersen, Investigation, Visualization, Methodology; Rosalina Fonseca, Formal analysis, Investigation, Methodology; Marco Capogna, Supervision, Investigation; Sadegh Nabavi, Conceptualization, Data curation, Formal analysis, Supervision, Funding acquisition, Validation, Investigation, Writing – original draft, Project administration, Writing - review and editing

### Author ORCIDs

Valentina Khalil ⬤ http://orcid.org/0000-0001-5928-8596
Sadegh Nabavi ⬤ http://orcid.org/0000-0002-3940-1210

### Ethics

All procedures that involved the use of mice were approved by the Danish Animal Experiment Inspectorate. (permit numbers: 2020-15-0201-00421 and 2023-15-0201-01431).

Reviewer #1 (Public review): https://doi.org/10.7554/eLife.91421.3.sa1
Reviewer #2 (Public review): https://doi.org/10.7554/eLife.91421.3.sa2
Author response https://doi.org/10.7554/eLife.91421.3.sa3

## Additional files

### Supplementary files

• MDAR checklist

### Data availability

The datasets used and/or analysed during the current study are available at the following link: https://github.com/NabaviLab-Git/Non-Hebbian-plasticity-transforms-transient-experiences-into-lasting-memories_Raw-data/tree/main.

The following dataset was generated:

| Author(s) | Year | Dataset title | Dataset URL | Database and Identifier |
|---|---|---|---|---|
| NabaviLab-Git | 2024 | Non-Hebbian-plasticity-transforms-transient-experiences-into-lasting-memories_Raw-data | https://github.com/NabaviLab-Git/Non-Hebbian-plasticity-transforms-transient-experiences-into-lasting-memories_Raw-data | GitHub, Non-Hebbian-plasticity-transforms-transient-experiences-into-lasting-memories_Raw-data |

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
