## [Editor Report · eLife assessment]

This study presents **important** novel findings on how heterosynaptic plasticity can transform a weak associative memory into a stronger one, or produce a memory when stimuli were not paired. This work expands our views on the role of temporal- and input-specific plasticity in shaping learning and memory processes. The evidence, based on state-of-the-art in vivo manipulations, activity recordings, and behavioral analysis, is **convincing**. Findings will be of broad interest to neuroscience community, and especially those studying synaptic plasticity and associative memory.

---

## [Referee Report · Reviewer #1 (Public review)]

Summary:

The authors goal here was to explore how a non hebbian form of plasticity, heterosynaptic LTP, could shape neuronal responses and learning. They used several conceptually and technically innovative approaches to answer this. First, they identified a behavioral paradigm that was a subthreshold training paradigm (stimulation of thalamic inputs with a footshock), which could be 'converted' to a memory via homosynaptic LTP (HFS of thalamic inputs). They then find that stimulation of 'cortical' inputs could also convert the subthreshold stimulation to a lasting memory, and that this was associated with a change in neuronal response, akin to LTP. Finally, they provide some slice work which demonstrated that stimulation of cortical inputs could stabilize LTP at thalamic inputs.

Strengths:

(1) The approach was innovative and asked an important question in the field.

(2) The studies are, for the most part, quite rigorous, using a novel dual opsin approach to probe multiple inputs in vivo.

(3) The authors explore neural responses both in vivo and ex vivo, as well as leveraging a 'simple' behavior output of freezing.

---

## [Referee Report · Reviewer #2 (Public review)]

Summary

Faress et al. address how synaptic plasticity (i.e. potentiation induced by high frequency stimulation, HFS) induced at different time points and pathways relative to those active during initial learning can transform memories. They adopt an experimental design developed by Nabavi et al, 2014 to optogenetically induce a weak fear memory by pairing an optical conditioned stimulus (CS) at thalamo-LA synapses with a footshock unconditioned stimulus (US) in male mice. Homosynaptic HFS delivered in the same pathway before or after conditioning transforms the weak memory into a stronger one. Leveraging a new dual wavelength optogenetic approach in vivo, they also show that heterosynaptic (cortico-LA) HFS directly following the opto-conditioning can transform the thalamo-LA induced fear memory, or create a memory when directly delivered after unpaired conditioning. Lastly, they demonstrate that heterosynaptic potentiation of the thalamo-LA pathway accompanies the strengthening of fear memory in freely moving mice. The authors conclude that a transient experience (i.e. weak memory) can be transformed into a stable one by non-Hebbian forms of plasticity.

Strengths

This study uses well-defined and elegant optogenetic manipulations of distinct neural pathways in awake behaving mice combined with in vivo recordings, which allows to directly manipulate and monitor synaptic strength and memory. It addresses an interesting, timely, and important question.

Weaknesses

A key experiment with in vivo monitoring of LFPs and behavior (Fig. 5a-c) seems a bit underpowered and input-output curves (extended data 5c) not entirely convincing.

Ex vivo slice experiments (Fig. 5d-f) are not well aligned with in vivo experimental conditions. While they provide proof of principle, this is not entirely novel (see Fonseca et al, 2013).

Significance and impact

The conclusions are well supported by the data. The significance of the study lies in showing in vivo, that plasticity induced at different times or synaptic pathways than those engaged during learning can modify a memory and the synaptic strength in the neural pathway related to that memory. While heterosynaptic and timing-dependent effects in synaptic plasticity have been described largely ex vivo on shorter time scales, the discovery of lasting behavioral effects on memory is novel. The study was enabled by a combination of clever approaches: creation of a "synthetic" pathway-specific association and a novel dual opsin approach in vivo to probe the role of plasticity in a converging second pathway at the same time.

This work broadens our understanding of how Hebbian and non-Hebbian forms of plasticity shape neural activity and associative memory and is of broad interest to the neuroscience community.

---

## [Author Response]

The following is the authors’ response to the original reviews.

**Reviewer #1 (Public Review):**
(1) There appears to be a flaw in the exploration of cortical inputs. the authors never show that HFS of cortical inputs has no effect in the absence of thalamic stimulation. It appears that there is a citation showing this, but I think it would be important to show this in this study as well.

We understand that the reviewer would like us to induce an HFS protocol on cortical input and then test if there is any change in synaptic strength in thalamic input. We have done this experiment which shows that without a footshock, high-frequency stimulation (HFS) of the cortical inputs did not induce synaptic potentiation on the thalamic pathway (Extended Data Fig. 4d).

(2) t is somewhat confusing that the authors refer to the cortical input as driving heterosynaptic LTP, but this is not shown until Figure 4J, that after non-associative conditioning (unpaired shock and tone) HFS of the cortex can drive freezing and heterosynaptic LTP of thalamic inputs.

We agree with the reviewer that it is in figure 4j and figure 5,b,c which we show electrophysiological evidence for cortical input driving heterosynaptic LTP. It is only to be consistent with our terminology that initially we used behavioral evidence as the proxy for heteroLTP (figure 3c).

…, the authors are 'surprised' by this outcome, which appears to be what they predict.

We removed the phrase “To our surprise”.

(3) 'Cortex' as a stimulation site is vague. The authors have coordinates they used, it is unclear why they are not using standard anatomical nomenclature.

We replaced “cortex” with “auditory/associative cortex”.

(4) The authors' repeated use of homoLTP and heteroLTP to define the input that is being stimulated makes it challenging to understand the experimental detail. While I appreciate this is part of the goal, more descriptive words such as 'thalamic' and 'cortical' would make this much easier to understand.

We agree with the reviewer that a phrase such as “an LTP protocol on thalamic and cortical inputs” would be more descriptive. We chose the words “homoLTP” and “heteroLTP” only to clarify (for the readers) the physiological relevance of these protocols. We thought by using “thalamic” and “cortical” readers may miss this point. However, when for the first time we introduce the words “homoLTP” and “heteroLTP”, we describe which stimulated pathway each refers to.

**Reviewer #2 (Public Review):**
(1) …The experimental schemes in Figs. 1 and 3 (and Fig. 4e and extended data 4a,b) show that one group of animals was subjected to retrieval in the test context at 24 h, then received HFS, which was then followed by a second retrieval session. With this design, it remains unclear what the HFS impacts when it is delivered between these two 24 h memory retrieval sessions.

We understand that the reviewer has raised the concern that the increase in freezing we observed after the HFS protocol (ex. Fig. 1b, the bar labeled as Wth+24hHFSth) could be caused or modulated by the recall prior to the HFS (Fig. 1a, top branch). To address this concern, in a new group of mice, 24 hours after weak conditioning, we induced the HFS protocol, followed by testing (that is, no testing prior to the HFS protocol). We observed that homoLTP was as effective in mice that were tested prior to the induction protocol as those that were not (Fig. 1b, Extended Data Fig. 1d,e).

It would be nice to see these data parsed out in a clean experimental design for all experiments (in Figs 1, 3, and 4), that means 4 groups with different treatments that are all tested only once at 24 h, and the appropriate statistical tests (ANOVA). This would also avoid repeating data in different panels for different pairwise comparisons (Fig 1, Fig 3, Fig 4, and extended Fig 4).

While we understand the benefit of the reviewer’s suggestion, the current presentation of the data was done to match the flow of the text and the delivery of the information throughout the manuscript. We think it is unlikely that the retrieval test prior to the HFS impacts its effectiveness, as confirmed by homosynaptic HFS data (Extended Data Fig. 1d,e). It is beyond the scope of current manuscript to investigate the mechanisms and manipulations related to reconsolidation and retrieval effects.

(2) … It would be critical to know if LFPs change over 24 h in animals in which memory is not altered by HFS, and to see correlations between memory performance and LFP changes, as two animals displayed low freezing levels. … They would suggest that thalamo-LA potentiation occurs directly after learning+HFS (which could be tested) and is maintained over 24 h.

We have performed the experiment where we recorded the evoked LFP 2hrs and 24hrs following the weak conditioning protocol. We observed that a weak conditioning protocol that was not followed by an optical LTP protocol on the cortical inputs failed to produce synaptic potentiation of the thalamic inputs (tested 2hrs and 24hrs after the LTP protocol; Extended Data Fig. 5d,e).

(3) The statistical analyses need to be clarified. All statements should be supported with statistical testing (e.g. extended data 5c, pg 7 stats are missing). The specific tests should be clearly stated throughout. For ANOVAs, the post-hoc tests and their outcomes should be stated. In some cases, 2-way ANOVAs were performed, but it seems there is only one independent variable, calling for one-way ANOVA.

All the statistical analyses have been revised and the post-hoc tests performed after the ANOVAs are mentioned in the relevant figure legends.

**Reviewer #2 (Recommendations For The Authors):**
The wording "transient" and "persistent" used here in the context of memory seems a bit misleading, as only one timepoint was assessed for memory recall (24 h), at which the memory strength (freezing levels) seem to change.

As the reviewer mentioned, we have tested memory recall only at one time point. For this reason, throughout the text we used “transient” exclusively to refer to the experience (receiving footshock) and not to the memory. We replaced “persistence” with “stabilization” where it refers to a memory (“the induction of plasticity influences the stabilization of the memory”).

For the procedures in which the CS and US were not paired, the term "unpairing" is used (which is probably the more adequate one), but the term "non-associative conditioning" appears in the text, which seems a bit misleading, as this term may have another connotation. There is also literature that an unpairing of CS and US could lead to the formation of a safety memory to the CS, that may be disrupted by HFS stimulation.

We replaced "non-associative" with “unpaired”.

Validation of viral injection sites for all experiments: Only representative examples are shown, it would be nice to see all viral expression sites.

For this manuscript, we have used 155 mice. For this reason, including the injection sites for all the animals in the manuscript is not feasible. Except for the mice that have been excluded, (please see exclusion criteria added in the methods), the expression pattern we observed was consistent across animals and therefore the images shown are true representatives.

Extended Data 1b: Please explain what N, U, W, and S behavioral groups mean. To what groups mentioned in the text (pg 2,3) do these correspond?

The requested clarifications are implemented in the figure legend.

Please elaborate on the following aspects of your methods and approaches:Please explain if the protocol for HFS to manipulate behavior was the same as the one used for the LTP experiments (Fig 1d, Fig 4j) and was identical for homo/hetero inputs from thal and ctx?

We used the same HFS protocol for all the HFS inductions. We included this information in the methods section.

Please state when the HFS was given in respect to the conditioning (what means immediately before and after?) and in which context it was given. Were animals subjected to HFS exposed to the context longer (either before or after the conditioning while receiving HFS) than the other groups? When the HFS was given in another context (for the 24 h group)- how was this controlled for?

Requested information has been added to the methods section. The control and intervention groups were treated in the same way.

When were the footshocks given in the anesthesized recordings (Fig. 4j) and how was the temporal relationship to the HFS? Was the timing the same as for the HFS in the behavioral experiments?

Requested information has been added to the methods section.

Please add information on how the LFP was stimulated and how the LFP- EPSP slope was determined in in vivo recordings, likewise for the whole cell recordings of EPSPs in Fig. 5d-f.

Requested information has been added to the methods section.

Here, the y-Axis in Fig. 5e should be corrected to EPSP slope rather than fEPSP slope if these are whole-cell recordings.

This has been corrected.

Please include information if the viral injections and opto-manipulations were done bilateral or unilateral and if so in which hemisphere. Likewise, indicate where the LFP recordings were done.

Requested information has been added to the methods section.

Were there any exclusion criteria for animals (e.g. insufficient viral targeting or placement of fibers and electrodes), other than the testing of the optical CS for adverse effects?

Requested information has been added to the methods section.

Statistics: In addition to clarifying analytical statistics, please clarify n-numbers for slice recordings (number of animals, number of slices, and number of cells if applicable).

Requested information has been added to the methods section.

It would be nice to scrutinize the results in extended data 4b. The freezing levels with U+24h HFS show a strong trend towards an increase, the effect size may be similar to immediate HFS Fig 4f and extended data 4a if n was increased.

We agree with the reviewer. To address this point, we added “HomoLTP protocol when delivered 24hrs later, produced an increase in freezing; however, the value was not statistically significant.” To show this point, we used the same scale for freezing in Extended Data Fig. 4a and b.

In the final experiment (Fig. 5a-c), Fig. 5b seems to show results from only one animal, but behavioral results are from 4 animals (Fig 5c). It would be helpful to see the quantification of potentiation in each animal.

The results (now with error bar) include all mice.

Please spell out the abbreviation "STC".

Now, it is spelled out.

Page 8 last sentence of the discussion does not seem to fit there.

The sentence has been removed.

**Reviewer #3 (Recommendations For The Authors):**
(1) The authors did not determine how WTh affects Th-LA synapses, as field EPSPs were recorded only after HFS. WTh was required for the effects of HFS, as HFS alone did not produce CR in naïve and/or unpaired controls. As such the effects of the WTh protocol on synaptic strength must be investigated.

We have performed the experiment where we recorded the evoked LFP 2hrs and 24hrs following the weak conditioning protocol. We observed that a weak conditioning protocol that was not followed by an optical LTP protocol on the cortical inputs failed to produce synaptic potentiation of the thalamic inputs (tested 2hrs and 24hrs after the LTP protocol; Extended Data Fig. 5d,e).

(2) The authors provide some evidence that their dual opsin approach is feasible, particularly the use of sustained yellow light to block the effects of blue light on ChrimsonR. However, this validation was done using single pulses making it difficult to assess the effect of this protocol on Th input when HFS was used. Without strong evidence that the optogenetic methods used here are fault-proof, the main conclusions of this study are compromised. Why did the authors not use a protocol in which fibers were placed directly in the Ctx and Th while using soma-restricted opsins to avoid cross-contamination?

We understand that the reviewer raises the possibility that our dual-opsin approach, although effective with single pulses, may fail in higher frequency stimulation protocols (10Hz and 85Hz). To address this concern, in a new group of mice we applied our approach to 10Hz and 85Hz stimulation protocols. We show that our approach is effective in single-pulse as well as in 10Hz and 85Hz stimulation protocols (Fig. 2d-h).